# Magnetic Structure and Origin of Insulating Behavior in the Ba_2_CuOsO_6_ System, and the Role of A-Site Ionic Size in Its Bandgap Opening: Density Functional Theory Approaches

**DOI:** 10.3390/nano12010144

**Published:** 2021-12-31

**Authors:** Taesu Park, Wang Ro Lee, Won-Joon Son, Ji-Hoon Shim, Changhoon Lee

**Affiliations:** 1Department of Chemistry, Pohang University of Science and Technology, Pohang 37673, Korea; tsalpha@postech.ac.kr; 2Faculty of Liberal Education, Jeonbuk National University, Jeonju 54896, Korea; wrlee@jbnu.ac.kr; 3Samsung Advanced Institute of Technology (SAIT), Samsung Electronics, 130 Samsung-ro, Yeongtong-gu, Suwon 16678, Korea; wonjoon.son@samsung.com; 4Division of Advanced Materials Science, Pohang University of Science and Technology, Pohang 37673, Korea; 5Max Planck POSTECH Center for Complex Phase of Materials, Pohang University of Science and Technology, Pohang 37673, Korea

**Keywords:** magnetic structure, BaOsCuO_6_, SOC effect, magnetic insulating behavior, orbital interaction

## Abstract

The magnetic structure and the origin of band gap opening for Ba_2_CuOsO_6_ were investigated by exploring the spin exchange interactions and employing the spin–orbit coupling effect. It revealed that the double-perovskite Ba_2_CuOsO_6_, composed of the 3d (Cu^2+^) and 5d (Os^6+^) transition metal magnetic ions is magnetic insulator. The magnetic susceptibilities of Ba_2_CuOsO_6_ obey the Curie–Weiss law, with an estimated Weiss temperature of −13.3 K, indicating AFM ordering. From the density functional theory approach, it is demonstrated that the spin exchange interaction between Cu ions plays a major role in exhibiting an antiferromagnetic behavior in the Ba_2_CuOsO_6_ system. An important factor to understand regarding the insulating behavior on Ba_2_CuOsO_6_ is the structural distortion shape of OsO_6_ octahedron, which should be closely connected with the ionic size of the A-site ion. Since the d-block of Os^6+^ (d^2^) ions of Ba_2_CuOsO_6_ is split into four states (xy < xz, yz < x^2^–y^2^ < z^2^), the crucial key is separation of doubly degenerated xz and yz levels to describe the magnetic insulating states of Ba_2_CuOsO_6_. By orbital symmetry breaking, caused by the spin–orbit coupling, the t_2g_ level of Os^6+^ (d^2^) ions is separated into three sublevels. Two electrons of Os^6+^ (d^2^) ions occupy two levels of the three spin–orbit-coupled levels. Since Ba_2_CuOsO_6_ is a strongly correlated system, and the Os atom belongs to the heavy element group, one speculates that it is necessary to take into account both electron correlation and the spin–orbit coupling effect in describing the magnetic insulating states of Ba_2_CuOsO_6_.

## 1. Introduction

Various osmium oxide compounds exhibit attractive magnetic and electronic phenomena developed from their electron correlation effects, such as the ferromagnetic gapped state in Ba_2_NiOsO_6_ [1], the singlet ground-state excitonic magnetism in Y_2_OsO_7_ [2,3,4], the spin-driven metal to insulator transition in Pb_2_CaOsO_6_ [5], and the unusual superconductivity in AOsO_6_ (A = Cs, Rb, and K) [6].

The solid-state osmium oxides are noteworthy for the following two reasons. One lies in understanding the origin of the band gap inducing of the metal to insulator transition. So far, various mechanisms for explaining the band gap opening for solid-state osmium oxide compounds have been extensively considered, such as the Mott-type mechanism, the d-level splitting pattern caused by the electron correlation effect, the Slater-type mechanism, and the orbital symmetry breaking mechanism, driven by the spin–orbit coupling (SOC) effect. As an example, the band gap opening for Cd_2_Os_2_O_7_ and NaOsO_3_ is explained by Slater-type insulators, which are associated with magnetic ordering [7,8,9].

For Ba_2_NaOsO_6_, there has been a debate on whether it is a Mott-type insulator or a SOC effect-driven insulator. In a study by Erickson et al., the nature of the insulating phase for Ba_2_NaOsO_6_ is represented as a Mott-type insulator [10,11]. Xiang et al. suggest that the insulating behavior of Ba_2_NaOsO_6_ should be developed by the simultaneous effects of electron correlation and SOC [12]. Although it is well known that the 5d-block element has considerably extended valence orbitals, resulting in weak on-site Coulomb repulsion, the magnetic insulating features of Sr_2_MOsO_6_ (M = Cu and Ni) are reproduced with a significantly large on-site repulsion at the Os atom site [13]. Thus, it is of great importance to explore the origin of the band gap opening in solid-state osmium oxide compounds.

The other issue on solid-state osmium oxide is related to the various oxidation states of the Os ion. Osmium forms compounds with oxidation states, ranging from −2 to +8. As an example, the oxidation states of the Os ion in Na_2_[Os(CO)], Na_2_[Os_4_(CO)_13_], Os_3_(CO)_12_, OsI, OsI_2_, OsBr_3_, OsO_2_, OsF_5_, OsF_6_, OsOF_5_, and OsO_4_ are −2, −1, 0, +1, +2, +3, +4, +5, +6, +7, and +8, respectively. A large spatial extension of Os at the 5d level is a main reason for the wide spectrum of oxidation states of the Os atom in osmium compounds. In general, the nd orbital of metal in the zero-oxidation state shows a spatial extension that increases in the order 3d < 4d < 5d, so that the widths of the d-block bands should increase in the order 3d < 4d < 5d. For this reason, 5d oxides show various valence states with the wide band widths of d-blocks. The electron correlation effects for 4d and 5d systems are weak while the effects of SOC are strong [14].

Recently, a new double perovskite osmium oxide Ba_2_CuOsO_6_ was synthesized under somewhat extreme conditions (~6 GPa and ~1800 K). The crystal structure and magnetic properties of Ba_2_CuOsO_6_ were characterized with synchrotron X-ray diffraction, thermo-gravimetric analysis, magnetic susceptibility, isothermal magnetization, and specific heat measurements [15]. The temperature dependence of the specific heat showed an electrically insulating behavior at all measured temperatures [15]. They also found that the Ba_2_CuOsO_6_ obeys the Curie–Weiss law with the estimated Weiss temperature −13.3 K [15]. Interestingly, a magnetic susceptibility measurement shows two T_max_ at ~55 K and ~70 K. This would be associated with two different magnetic ions in Ba_2_CuOsO_6_, which lead to more than two types of magnetic sublattice.

The Ba_2_CuOsO_6_ crystallizes in a tetragonal space group, I4/m, in which the valence states of the Os and Cu atoms are Os^6+^ (d^2^, S = 1) and Cu^2+^ (d^9^, S = 1/2), respectively. The Cu^2+^ and Os^6+^ ions are located on the perovskite B-site, and they form the CuO_6_ and OsO_6_ octahedrons, respectively. They share their corners in all crystallographic directions, such that alternating CuO_6_ and OsO_6_ run in all three crystallographic directions, as shown in Figure 1. The Ba^2+^ ion is located on the center of the Cu_4_Os_4_ cube (see Figure 1). With the structural distortion of CuO_6_ and OsO_6_ octahedrons from the ideal MO_6_ octahedron, the Cu-O-Os bridges in the ab plane are bent, whereas those along the c direction are linear, which should be closely related with the orbital splitting and the orbital occupancy.

One can easily predict that the axial Cu-O bond is elongated by a strong Jahn–Teller distortion, associated with the electron configuration of the Cu^2^^+^ (d^9^) ion in each CuO_6_ octahedron, with two long Cu-O_ax_ bonds along the z^2^ orbital direction (crystallographic c direction) and four short equatorial Cu-O bonds in the x^2^–y^2^ orbital locating plane (the crystallographic ab plane). Besides, in each OsO_6_ octahedron, the (t_2g_)^2^ electron configuration of the Os^6^^+^ ion exhibits a weak Jahn–Teller distortion, associated with spatial extension of 5d orbital (see Figure 1). As a consequence, unlike CuO_6_ octahedrons, each OsO_6_ octahedron is weakly compressed in the axial direction with two short Os-O_ax_ bonds along the c direction and four elongated Os-O_eq_ bonds in the ab plane.

Here, we examined the causes of several interesting and seemingly puzzling phenomena, such as the magnetic structures and the origin of the insulating phase in Ba_2_CuOsO_6_, by performing the DFT, the DFT + U, and the DFT + U + SOC electronic band structure calculations. Then, we studied the spin exchange interactions of Ba_2_CuOsO_6_ by a relative energy-mapping analysis. The magnetic properties of Ba_2_CuOsO_6_ were explained by the aspect of their orbital interactions and their spin exchange interactions. The double antiferromagnetic (AFM) anomaly at ~55 K and ~70 K of the susceptibility curve was investigated by analyzing its spin exchange interaction. Finally, we examined the reason for the insulating phase on the basis of the DFT studies and a perturbation theory analysis using ${\hat{H}}_{soc}$ as perturbation.

## 2. Computational Details

In our DFT calculations, we employed the frozen-core, projector-augmented wave method [16,17], encoded in the Vienna ab initio simulation package (VASP) [18], and the generalized-gradient approximation of Perdew, Burke, and Ernzerhof [19], for the exchange correlation functional, with the plane-wave cut-off energy of 450 eV and a set of 48 k-points for the irreducible Brillouin zone.

In general, the orbitals of the 5d element are much more diffuse than 3d orbitals, so the U_Os_ value is expected to be smaller than the U value of Cu. However, in Ba_2_CuOsO_6_, the oxidation state of Os ion is +6, so that 5d state of Os^6+^ ion is strongly contracted. Note that the ionic size of six coordinated Cu^2+^ ions (0.870 Å) is larger than that of six coordinated Os^6+^ ions (0.685 Å) [20]. Moreover, the electronegativity of six coordinated Os^6+^ and Cu^2+^ ions are 2.362 and 1.372, respectively [21]. Thus, we use the larger U values on the Os^6+^ ion than on the Cu^2+^ ion. In our DFT calculation, we employed U value sets of U_Cu_ = 2, U_Os_ = 3 eV and U_Cu_ = 4, U_Os_ = 5 eV, respectively [22].

The measured electrical resistivity (ρ) vs. temperature of Ba_2_CuOsO_6_, given in the previous study [15], shows an insulating behavior for all temperature ranges, which means that the insulating behavior of Ba_2_CuOsO_6_ is an intrinsic nature, not a phenomenon coupled with magnetic ordering. Additionally, Khaliullin et al. suggest that the band gap is developed from excitonic transition from singlet to triplet in the Y_2_Os_2_O_7_ system [4]. The Y_2_Os_2_O_7_ (Os^4+^, d^4^) belongs to the van Vleck-type Mott system, showing non-magnetic ground state, due to SOC and weak temperature dependence, such as uniform magnetic susceptibility above Neel temperature. However, magnetic susceptibility curve in Ba_2_CuOsO_6_ showed Curie–Weiss behavior over a broad temperature range [15], implying that it is not a case of suppressed effective magnetic moment by SOC. Thus, we considered the ferromagnetic (FM) state of Ba_2_CuOsO_6_ in our DFT + U + SOC calculation, even though this system undergoes the antiferromagnetic ordering. In order to find proper U value set for reproducing the insulating state of Ba_2_CuOsO_6_, we perform systematic DFT + U + SOC calculations with various U value sets. However, all U sets failed to reproduce the insulating state of Ba_2_CuOsO_6_ except U value set of U_Cu_ = 4 and U_Os_ = 5 eV (see Appendix A). Thereby, we carried out the DFT + U + SOC calculation with the U value set of U_Os_ = 5 and U_Cu_ = 4 eV for understanding the origin of the insulating behavior on Ba_2_CuOsO_6_.

## 3. Spin Exchange Interaction and Spin Lattice

In understanding the magnetic structure of Ba_2_CuOsO_6_, it is important to inspect its local structure and its following spin exchange paths. Figure 2 shows eight possible spin exchange paths of Ba_2_CuOsO_6_. There are three spin exchanges between Cu ions, three spin exchanges between Os ions, and two Cu-Os spin exchanges. The J_1_ and J_2_ are the superexchange (SE) type, involving the Cu-O-Os path, while the J_3_–J_8_ are super-superexchange (SSE), involving M-O···M-O exchange path (M = Cu or Os). The J_1_, J_3_, J_4_ J_7_, and J_8_ exchanges are ab plane interactions and the J_2_, J_5_, and J_6_ exchanges are interactions between ab planes. The geometrical parameters associated with these paths are listed in Table 1.

Let us examine the relationship between the spin exchange interactions J_1_–J_8_ and the structural parameters associated with their exchange pathways and the spin exchange interactions in a viewpoint of orbital interaction. As mentioned, the CuO_6_ and OsO_6_ octahedrons are distorted from regular MO_6_ octahedrons. The nature of distortion should be explained by the crystal field effect, the Jahn–Teller instability, and the different magnitude of interaction between the Ba^2+^ ion and the MO_6_ (M = Os, and Cu) octahedron. The distortion of the CuO_6_ and OsO_6_ octahedrons plays an important role in the nature and the strength of the spin exchanges. The CuO_6_ octahedron is axially elongated and the OsO_6_ octahedron is axially shrunk by the Jahn–Teller instability. Thus, the d-orbital sequence of Cu^2+^ ion is xz, yz < xy < z^2^ < x^2^–y^2^ and the d-orbital splitting of Os^6+^ ion is xy < xz,yz < x^2^–y^2^ < z^2^. As depicted in Figure 3a,b, the magnetic orbitals of Cu^2+^ ion and Os^6+^ are singly occupied x^2^–y^2^ orbital and two orbitals of three t_2g_ orbitals, respectively. The structural parameters are listed in Table 1. The ∠Cu-O-Os angle in J_1_ path is close to 180° (172.6°), so the J_1_ exchange should be strong FM, because of orthogonality between Cu x^2^–y^2^ and Os t_2g_ orbitals, according to Goodenough’s role [23]. In the J_2_ exchange, the Cu-O and Os-O bonds of the Cu-O-Os linkage does not contain magnetic orbitals and the ∠Cu-O-Os angle is 180°. Thus, the Cu^2+^ ion and the Os^6+^ ion in the J_2_ exchange do not interact. The J_2_ exchange interaction should be a very weak interaction. The J_3_ exchange is unsymmetrical and the ∠Cu-O···O-Cu angles are 127.6 and 142.4°, which contain magnetic orbitals, indicating possible good orbital overlap between Cu^2+^ ions (see Figure 3d). Thus, we expect that J_3_ should be AFM. The geometrical structure of J_4_ exchange is almost identical to that of the J_3_ exchange. However, the J_4_ exchange contains two magnetic orbitals, indicating multiple channel spin exchange interactions, suggesting a presence of a strong orbital overlap via the Os-O···O-Os linkage (Figure 2a). The J_4_ spin exchange should have strong AFM because the two magnetic orbitals of the t_2g_ orbitals can overlap across their O···O contacts, as shown in Figure 3e. The J_5_ and J_6_ exchanges are the interactions between the adjacent ab layers, and the O···O contact distances of the SSE path J_5_ and J_6_ are 4.264 Å. The M-O bonds of the M-O···O-M linkage are not located in the magnetic orbital plane. One can predict that the magnetic interaction of J_5_ and J_6_ exchange path is negligibly weak.

The ∠M-O···O-M angles of the SE paths J_7_ and J_8_ are close to 180° (172.6°) and the M-O bond of M-O···M-O linkage contains a magnetic orbital. Thus, the spin exchange interaction J_7_ and J_8_ should have strong AFM. However, the magnitude of the J_7_ spin exchange is much stronger than that of J_8_. It is due to the fact that the orbital interaction in the J_7_ exchange is a π-type orbital interaction, while the orbital interaction in J_8_ is a σ-type orbital interaction, as shown in Figure 3f,g.

We extracted the spin exchange interactions using the DFT + U + SOC calculation to elucidate our analysis of the spin exchange interactions in terms of the local geometrical structure and the orbital interaction analysis. To extract the values for the J_1_–J_8_ exchanges, we carried out a total energy calculation of the nine ordered spin configurations of Ba_2_CuOsO_6_, shown in Figure 4. To obtain the values of J_1_–J_8_, we determined the relative energies of the abovementioned cases obtained from the DFT + U + SOC calculations with U_Cu_ = 2, U_Os_ = 3 eV and U_Cu_ = 4, U_Os_ = 5 eV. The relative energies of various AFM cases obtained are summarized in Figure 4. In terms of the spin Hamiltonian $\hat{H}=-\sum_{i<j}J_{\mathrm{ij}}{\hat{S}}_{i}\cdot {\hat{S}}_{j}$, where J_ij_ = J_1_–J_8_, the total spin exchange energies of these states per formula units (FUs) are expressed as follows:E_spin_ = (n_1_J_1_)(MN/4) + n_2_J_2_(MN/4) + n_3_J_3_(M^2^/4) + n_4_J_4_(N^2^/4) + n_5_J_5_(M^2^/4) + n_6_J_6_(N^2^/4) + n_7_J_7_(M^2^/4) + n_8_J_8_(N^2^/4) (1)
by using the energy expressions obtained for spin dimers with M and N unpaired spins per spin site (here, M(Cu^2+^) = 1 and N(Os^6+^) = 2) [24]. The values of n_1_−n_8_ for the eight ordered spin states, FM, and AF1–AF8, are described in Appendix A.

Thus, by mapping the relative energies of the nine spin arrangements presented in Figure 2 onto corresponding energies expected from Equation (1), we obtain J_1_–J_8_, as summarized in Table 2.

The determined spin exchange interactions, based on the DFT + U + SOC calculation (U = 4 and 5 eV for Cu and Os), show that the relative strength of the spin exchange interactions associated with their strongest interaction, J_7_, are |J_1_| ≈ 0.385 |J_7_|, |J_2_| ≈ 0.005 |J_7_|, |J_3_| ≈ 0.283 |J_7_|, |J_4_| ≈ 0.850 |J_7_|, |J_5_| ≈ 0.015 |J_7_|, |J_6_| ≈ 0.025 |J_7_|, and |J_8_| ≈ 0.339|J_7_|, respectively. This suggests that the magnetic property of Ba_2_CuOsO_6_ is mainly governed by J_1_, J_3_, J_4_, J_7_, and J_8_ exchanges, in which all of them are AFM except J_1_. The J_1_ exchange is an FM interaction, as expected in the previous section. The J_4_ and J_7_ spin exchanges are much stronger than others. As a consequence, the magnetic property of the Ba_2_CuOsO_6_ system mainly comes from the J_4_ and J_7_ spin exchange interactions, which would deduce the existence of magnetic sublattice. In addition, the observed phenomena of two T_max_ at ~55 K and ~70 K in susceptibility measurement would support such a possibility of magnetic sublattice [15]. The presence of two different magnetic ions in Ba_2_CuOsO_6_ system should be related with the double AFM-like anomaly at ~55 K and ~70 K. We roughly obtained Neel temperature T_N_ using a mean field approximation with the extracted spin exchange interactions, based on the DFT + U + SOC calculation (U_Cu_ = 4 and U_Os_ = 5 eV). The calculated Neel temperatures for the Cu^2+^ ion and Os^6+^ ion sublattices are 256 K and 297 K, respectively. The expected T_N_ for the Cu and Os sublattices is largely overestimated. The overestimation of the calculated T_N_ is comprehensible because it is well known that the DFT calculations generally overestimate the magnitude of spin exchange interactions by a factor of, approximately, up to four [25,26,27,28].

Although there is a possibility of spin frustration occurred by (J_1_,J_1_,J_7_), (J_1_,J_1_,J_8_), (J_1_,J_1_,J_3_), (J_1_,J_1_,J_4_), (J_3_,J_3_,J_7_), and (J_4_,J_4_,J_8_) triangles, it does not occur due to the fact that the strong J_4_ and J_7_ exchanges are forced to avoid spin frustration. This is in good agreement with the experimental result, in which the spin frustration factor (f = |θ|/|T_N_|) is just ~0.24, and there is no divergence between the zero-field and field susceptibility curves in Ba_2_CuOsO_6_ indicating the absence of spin frustration [15]. For a spin frustrated magnetic system, it is generally expected that the ratio, f = |θ|/|T_N_|, is greater than 6 [29,30,31]. The spin frustration factor f-value for Ba_2_CuOsO_6_ system is much lower than this critical value. Thus, the low f-value suggests that the spin frustration is very weak or does not occur in Ba_2_CuOsO_6_, despite its similar system (Sr_2_CuOsO_6_, Sr_2_CuIrO_6_) [32,33] showing strong spin frustration.

In summary, the overall magnetic property of the Ba_2_CuOsO_6_ system should be explained by AFM, and it does not show spin frustration.

## 4. Describing the Magnetic Insulating Behavior of Ba_2_CuOsO_6_

As mentioned in the introduction section, the Ba_2_CuOsO_6_ system is a magnetic insulator. Feng et al. [15] measured the temperature dependence of resistivity ρ of polycrystalline Ba_2_CuOsO_6_, in which it showed an insulating behavior at all measured temperature ranges. This reveals that the insulating behavior of Ba_2_CuOsO_6_ is an intrinsic nature, not a phenomenon coupled with magnetic ordering.

Although they mentioned the origin of the band gap in terms of a theoretical approach, they did not show evidence as well as a discussion for opening the band gap. Here, we discuss the origin of the insulating state of Ba_2_CuOsO_6_. The calculated electronic structures for Ba_2_CuOsO_6_ are presented in Figure 5 and Figure 6. The main distribution near the Fermi level comes from the Os^6+^ ion rather than the Cu^2+^ ion, which connotes that the Os^6+^ 5d states are mainly concerned with the insulating behavior of Ba_2_CuOsO_6_ (see Figure 5b,c).

For Sr_2_CuOsO_6_, it should be required to use larger Hubbard value on the Os atom than usual in realizing the magnetic insulating state [13]. Each OsO_6_ octahedron with axially elongated OsO_6_ octahedron (i.e., Os-O_ax_ = 1.928 (×2) Å, Os-O_eq_ = 1.888 (×4) Å) appears slightly distorted, compared with the CuO_6_ octahedron, caused by a weak Jahn–Teller instability. Therefore, the Os 5d state of axially elongated OsO_6_ octahedron is split into four states (xz,yz < xy < z^2^ < x^2^–y^2^) by the Jahn–Teller distortion. Two t_2g_ electrons in the Os^6+^ ion are now occupied to the degeneracy-lifted xz and yz states (see Figure 1). The band gap is then created by the energy difference between occupied (xz, yz) and unoccupied states (xy). The large Hubbard U value, which enhances the electron correlation effect, is enough to describe the insulating behavior of Sr_2_CuOsO_6_. On the other hand, each OsO_6_ of Ba_2_CuOsO_6_ has an axially shrunk octahedron, indicating the different types of Os 5d state splitting with Sr_2_CuOsO_6_. The Os 5d state of Ba_2_CuOsO_6_ is split into four states (xy < xz,yz < x^2^–y^2^ < z^2^). However, the Os 5d state splitting of Ba_2_CuOsO_6_ is very weak compared to that of Sr_2_CuOsO_6_, as shown in the Figure 1 and Figure 5c. Split Os 5d state of Ba_2_CuOsO_6_ is presented in Figure 5c, in terms of the projected DOS, which shows that the t_2g_ level of the Os 5d state is very weakly separated into the two states. In the (t_2g_)^2^ electron of the Os^6+^, one electron is occupied in the lowest xy state, and the remaining electron is occupied in doubly degenerated xz and yz states, as depicted in Figure 1. Thus, one can predict that the simple DFT and DFT + U approaches are insufficient to describe the magnetic insulating behavior of Ba_2_CuOsO_6_. The crucial key to describing the insulating behavior of Ba_2_CuOsO_6_ is in splitting the degenerate xz and yz states of the Os^6+^ ion. The SOC effect splits the t_2g_ state into three substates by the orbital symmetry breaking. Thus, one can speculate that the SOC effect should play an important role in describing the insulating behavior of Ba_2_CuOsO_6_, thereby it is necessary to consider the SOC effect.

Moreover, since the Os atom belongs to a heavy element group, the SOC effect should be expected to have a dramatically strong effect on the electronic structure of Ba_2_CuOsO_6_.

The spin–orbital part of the Hamiltonian in the Os sphere is then given by the following:(2)$${\hat{H}}_{SOC}=\lambda \hat{S}\cdot \hat{L}$$
where the SOC constant *λ* > 0 for the Os^6+^ (d^2^) ion, with less than half-filled t_2g_ levels. With *θ* and *φ* as the azimuthal and polar angles of the magnetization in the rectangular crystal coordinate system, respectively, the $\hat{L}$ and $\hat{S}$ terms are rewritten [34,35] as follows:(3)$${\hat{H}}_{soc}=\lambda {\hat{S}}_{z^{\prime }}\left({\hat{L}}_{z}\cos\theta +\frac{1}{2}{\hat{L}}_{+}e^{-i\phi }\sin\theta +\frac{1}{2}{\hat{L}}_{-}e^{i\phi }\sin\theta \right)\;+\frac{\lambda }{2}{\hat{S}}_{{+}^{\prime }}\left(-{\hat{L}}_{z}\sin\theta -{\hat{L}}_{+}e^{-i\phi }\sin^{2}\frac{\theta }{2}+{\hat{L}}_{-}e^{i\phi }\cos^{2}\frac{\theta }{2}\right)\;+\frac{\lambda }{2}{\hat{S}}_{{-}^{\prime }}\left(-{\hat{L}}_{z}\sin\theta +{\hat{L}}_{+}e^{-i\phi }\cos^{2}\frac{\theta }{2}-{\hat{L}}_{-}e^{i\phi }\sin^{2}\frac{\theta }{2}\right)$$

Since the spin-up and spin-down t_2g_ states are separated by the exchange splitting in the first order approximation, there is no need to consider interactions between different spin-up and -down states in SOC. Thus, one simply needs to consider only spin-up parts of t_2g_ states in using the degenerate perturbation theory, which requires calculation of the matrix elements $i|{\hat{H}}_{so}|j$ (*i*, *j* = *xy*, *yz*, *xz*) [6]_._ For that reason, only the *S_z_* operator term, as in the first line of Equation (3), brings about non-zero matrix elements. In evaluating these matrix elements, it is convenient to rewrite the angular parts of the *xy*, *yz*, and *xz* orbitals in terms of the spherical harmonics [12], as follows:(4)$$d_{xy}=\frac{-i}{\sqrt{2}}\left(Y_{2}^{2}-Y_{2}^{2}\right)\;d_{yz}=\frac{i}{\sqrt{2}}\left(Y_{2}^{1}+Y_{2}^{-1}\right)\;d_{xz}=\frac{-1}{\sqrt{2}}\left(Y_{2}^{1}-Y_{2}^{-1}\right)$$

Using these functions, the matrix representation elements $i|{\hat{H}}_{so}|j$ (*i*, *j* = *xy*, *yz*, *xz*) are found as follows:(5)iℏλ2(0sinθsinϕ−sinθsinϕ−sinθsinϕ0cosθsinθcosϕ−cosθ0)

By the diagonalization of Equation (5), we obtain eigenvalues of the three spin–orbit coupled states, $E_{1}=-\hbar \lambda /2$, $E_{2}=0$, and $E_{3}=\hbar \lambda /2$. The associated eigenfunctions $\psi_{1}$,$\;\psi_{2}$, and$\;\psi_{3}$ are given [12,34,35] by the following:(6)$$\begin{array}{l} \psi_{1}=\frac{\sqrt{2}}{2}[sin\theta d_{xy}+\left(isin\phi -cos\theta cos\phi \right)d_{yz}-\left(icos\phi +cos\theta sin\phi \right)d_{xz}]\\ \psi_{2}=\frac{\sqrt{2}}{2}[sin\theta d_{xy}-\left(isin\phi +cos\theta cos\phi \right)d_{yz}+\left(icos\phi -cos\theta sin\phi \right)d_{xz}]\\ \psi_{3}=cos\theta d_{xy}+sin\theta cos\phi d_{yz}+sin\theta sin\phi d_{xz} \end{array}$$

The above analysis indicates that the SOC effect splits the t_2g_ states into three substates by orbital symmetry breaking. For the Os^6+^ (d^2^) ion, two electrons of t_2g_ state should occupy $\psi_{1}\;$and$\;\psi_{2}$. Therefore, the band gap around the Fermi level should be developed in between the occupied $\psi_{2}$ and the unoccupied $\psi_{3}$ state.

To elucidate the above discussion, we carried out calculations to reproduce the insulating state of Ba_2_CuOsO_6_ by employing different theoretical methods, namely, simple DFT, DFT + U (U_Cu_ = 4, U_Os_ = 5 eV), and DFT + SOC methods, but all of them failed to reproduce the insulating behavior for Ba_2_CuOsO_6_ (see Figure 5a). Metallic electronic structures obtained from the DFT and the DFT + U calculations are already expected because the exchange splitting is not enough condition to open the band gap in Ba_2_CuOsO_6_. However, the electronic structure obtained from the DFT + SOC still failed to reproduce the insulating behavior for Ba_2_CuOsO_6_ (See Figure 5a). Presented in Figure 3a, the spin-up and spin-down bands are overlapped, which leads to a metallic state for Ba_2_CuOsO_6_. This means that not only the breaking of orbital symmetry by adapting the SOC effect, but also the separation of energy between the filled level (spin-up) and the empty level (spin-down) by increasing exchange splitting, should be required to describe the insulating state of Ba_2_CuOsO_6_. On-site repulsion, which properly describes the electron correlation effect, gives help to enhance the exchange splitting; thereby, the energy separation between the filled and the unfilled states within each spin channel is increased. Thus, the on-site repulsion should be also a crucial key to be considered in explaining the insulating state of Ba_2_CuOsO_6_.

To gain insight into this analysis, we carried out the band structure calculation, considering the electron correlation and the SOC effect simultaneously. The band structure calculated for the FM state of Ba_2_CuOsO_6_ using the DFT + U + SOC (U_eff_ = 4 and 5 eV on Cu and Os) is shown in Figure 6b. Indeed, the calculated electronic structure clearly shows an insulating band gap. In consequence, the interplay between the electron correlation and the SOC effect plays an essential role in opening a band gap for Ba_2_CuOsO_6_.

Meanwhile, there remains one question about the origin of the band gap opening in Ba_2_CuOsO_6_. The insulating behavior of Ba_2_CuOsO_6_ is intimately linked with the structural distortion of the OsO_6_ octahedron. We remind that the OsO_6_ octahedron for Ba_2_CuOsO_6_ is an axially compressed octahedral shape, while the OsO_6_ octahedron of Sr_2_CuOsO_6_ is an axially elongated octahedral shape. This implies that they undergo different types of structural distortion by the Jahn–Teller instability, which leads to different orbital splitting at the Os^6+^ t_2g_ level. Thus, they show different chemical and physical properties, especially in explaining the mechanism of band gap opening. The only difference between Sr_2_CuOsO_6_ and Ba_2_CuOsO_6_ is an ionic size of the A-site ion, namely, the ionic size of the Ba^2+^ ion is larger than that of the Sr^2+^ ion. In A_2_CuOsO_6_ (A = Sr and Ba), the A-site ion is located in the center of Cu_4_Os_4_ distorted cubes and distorted (O_ax_)_4_ squares, presented in Figure 1c. It is connected with 8 O_eq_ and 4 O_ax_ atoms to form a 12-coordinate AO_12_ (see Figure 1c). In A_2_CuOsO_6_, a change in the A-O_eq_ distance affects a, b, and c lattice parameters symmetrically; whereas, a change of A-O_ax_ distance affects the lattice parameter a (=b). Assuming that the increase of the A-O_ax_ distance leads to structural distortion forming axially shrunk type-OsO_6_ octahedrons, by increasing the Os-O_eq_ distance in the ab plane, while the decrease of the A-O_ax_ distance causes structural distortion to have axially elongated OsO_6_ octahedrons by decreasing the Os-O_eq_ distance. The A-O_ax_ distance should be mainly dominated by the ionic size of the A-site ion. Since the ionic size of the Ba^2+^ ion is larger than that of the Sr^2+^ ion, the Ba-O_ax_ distance is longer than the Sr-O_ax_ distance, which gives rise to a much longer Os-O_eq_ distance on Ba_2_CuOsO_6_, associated with the expansion of its lattice parameter a (=b). On the other hand, the effect of ionic size of the A-site on the change of the Os-O_ax_ distance is relatively insignificant. Indeed, the Os-O_eq_ distances in Ba_2_CuOsO_6_ and Sr_2_CuOsO_6_ are 1.960 and 1.888 Å, respectively, while the Os-O_ax_ distances in Ba_2_CuOsO_6_ and Sr_2_CuOsO_6_ are 1.928 Å and 1.946 Å, respectively [15,33]. Hence, the ionic size of the A-site ion should play an important role in determining the shape of the OsO_6_ octahedron in A_2_CuOsO_6_.

To verify the importance of the ionic size effect, we examine the aforementioned questions with the DFT + U + SOC (U_Cu_ = 4, U_Os_ = 5 eV) calculation for Ba_2_CuOsO_6_. We imagine the hypothetical compounds of A_2_CuOsO_6_ (A = Sr, Ca) by replacing the Ba atoms of Ba_2_CuOsO_6_ with other alkali earth atoms—Sr and Ca. A_2_CuOsO_6_ (A = Ba, Sr, and Ca) are fully optimized with the DFT + U + SOC (U_Cu_ = 4, U_Os_ = 5 eV) calculation. The optimized atomic positions and cell parameters are presented in the Appendix A. Results show that the lattice parameters decrease gradually with the decreasing ionic size of the A-site ion, but the decreasing of the a lattice parameter is greater than that of the c lattice parameter, which means that each Os-O_eq_ distance is increased as the A-site ionic size increases. Therefore, the cooperative effect of the A-site ionic size and the Jahn–Teller distortion, is responsible not only for the axially compressed OsO_6_ octahedrons in Ba_2_CuOsO_6_ but also for the axial elongation of OsO_6_ octahedrons in Sr_2_CuOsO_6_; this is closely related to the description of the insulating behavior of A_2_CuOsO_6_ (A = Ba, Sr, and Ca).

## 5. Concluding Remarks

The magnetic structure and the origin of the band gap opening for Ba_2_CuOsO_6_ are investigated by exploring the spin exchange interactions and analyzing the spin–orbit coupling effect. The magnetic property of Ba_2_CuOsO_6_ is explained by AFM, and it does not show spin frustration caused by the strong AFM interactions of J_4_ and J_7_. The structural distortion shape of the OsO_6_ octahedron, which should be closely connected with the ionic size of the A-site ion, is an important factor in understanding the insulating behavior of Ba_2_CuOsO_6_. Each OsO_6_ octahedron of Sr_2_CuOsO_6_ displays an axially elongated octahedral shape, while each OsO_6_ octahedron of Ba_2_CuOsO_6_ exhibits an axially compressed octahedral shape, which is caused by an ionic size effect of the A-site ion. Therefore, the t_2g_ level splitting of the Os^6+^ ions of Ba_2_CuOsO_6_ and Sr_2_CuOsO_6_ by the Jahn–Teller instability is differently depicted in Figure 1. Consequently, to explain the magnetic insulating states of Sr_2_CuOsO_6_, which are isostructural and isoelectronic in Ba_2_CuOsO_6_, it is necessary to properly employ an electron correlation effect. On the other hand, a cooperative effect of electron correlation and spin–orbit coupling is essential in describing the insulating behavior of Ba_2_CuOsO_6_, which is highly related with the t_2g_ orbital splitting of the Os^6+^ ion.

## Data Availability

All data are available on request from the corresponding author.

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
