# Peer review of "Magnetic Structure and Origin of Insulating Behavior in the Ba2CuOsO6 System, and the Role of A-Site Ionic Size in Its Bandgap Opening: Density Functional Theory Approaches"

_nanomaterials, 2021, doi:10.3390/nano12010144_

Round 1

Reviewer 1 Report

This manuscript by Lee et al. reports on theoretical investigation on the magnetic structure and origin of band gap opening for Ba2CuOsO6, the targeted compound was studied experimentally previously. The spin exchange interactions and spin-orbit coupling effect were analyzed, antiferromagnetic coupling is shown to dominate the magnetic property of Ba2CuOsO6. Structural distortion of OsO6 octahedron is revealed to be closely related to the insulating behavior of Ba2CuOsO6. In my opinion, the results reported here are interesting to the field. The story is well organized. I recommend publishing on the current shape.

Author Response

[1] Review 1

 “This manuscript by Lee et al. reports on theoretical investigation on the magnetic structure and origin of band gap opening for Ba2CuOsO6, the targeted compound was studied experimentally previously. The spin exchange interactions and spin-orbit coupling effect were analyzed, antiferromagnetic coupling is shown to dominate the magnetic property of Ba2CuOsO6. Structural distortion of OsO6 octahedron is revealed to be closely related to the insulating behavior of Ba2CuOsO6. In my opinion, the results reported here are interesting to the field. The story is well organized. I recommend publishing on the current shape.”

Response: Thank you for your fruitful and helpful comments and valuable suggestions sent to us for improving our manuscript.  We correct the typo and English language and style in the manuscript.

Reviewer 2 Report

The manuscript presents the mechanism of gap opening in Ba2CuOsO6 in the context of DFT approach. These results are rather insightful and the calculations are the-state-of-the art. I find the manuscript valuable and recommend publication after the extension of discussion indicated below.

The ionic configuration Os^{4+} is known to demonstrate the interplay between singlet and triplet ions. Only triplet states are discussed in the present manuscript. The manuscrpt would improve if the authors discussed the reason for that. Why the singlet states can be always neglected for the ground states in this compound? This discussion would be complete if the idea of excitonic mechanism was highlighted, see G. Khaliullin, PRL 111, 197201 (2013).

I recommend acceptance after this extension is made.

Author Response

[2] Review 2

 “The manuscript presents the mechanism of gap opening in Ba2CuOsO6 in the context of DFT approach. These results are rather insightful and the calculations are the-state-of-the art. I find the manuscript valuable and recommend publication after the extension of discussion indicated below. The ionic configuration Os^{4+} is known to demonstrate the interplay between singlet and triplet ions. Only triplet states are discussed in the present manuscript. The manuscript would improve if the authors discussed the reason for that. Why the singlet states can be always neglected for the ground states in this compound? This discussion would be complete if the idea of excitonic mechanism was highlighted, see G. Khaliullin, PRL 111, 197201 (2013). I recommend acceptance after this extension is made.”

Response : We are very much thankful to the learned and respected referee for his presented report, we have revised our manuscript accordingly as suggested by the respected referee.

The Curie-Weiss behavior of magnetic susceptibility curve over broad temperature range implies that ground state of Ba2CuOsO6 is far from nonmagnetic Van Vleck-type system showing weak temperature dependence such as uniform magnetic susceptibility in Ca2RuO4. In Van Vleck-type systems, ground state is nonmagnetic J = 0 singlet state with suppressed effective moment but magnetism is induced by excitonic transition from singlet to triplet. In this reason, we did not consider the nonmagnetic singlet ground state for Ba2CuOsO6. Therefore, we considered the ferromagnetic state (triplet state) of Ba2CuOsO6 in our DFT+U+SOC calculation even though this system undergoes the antiferromagnetic ordering.

In order to address reviewer comment we add the following discussion in the computational details part.

Also, Khaliullin et al suggest that the band gap is developed from excitonic transition from singlet to triplet in Y2Os2O7 system3. The Y2Os2O7 (Os4+ , d4) belongs to Van Vleck-type Mott system showing nonmagnetic ground state due to SOC and weak temperature dependence such as uniform magnetic susceptibility above Neel temperature. However, magnetic susceptibility curve in Ba2CuOsO6 showed Curie-Weiss behavior over broad temperature range11 implying that it is not the case of suppressed effective magnetic moment by SOC